Modulation of carbon-to-nitrogen ratio shapes the microbial ecology in a methanol-fed recirculating marine denitrifying reactor

Lestin Livie
Villemur Richard richard.villemur@inrs.ca
Centre Armand-Frappier Santé Biotechnologie, Institut National de la Recherche Scientifique , Laval , Québec , Canada
García-Contreras Rodolfo
Electronic publication date: 2025 Oct 13
Publication date: 2025
Volume: 13
Electronic Location ID: e20129
Received 2025 Apr 29; Accepted 2025 Sep 2
Copyright: ©2025 Lestin and Villemur
Copyright year: 2025
Copyright holder: Lestin and Villemur
License: This is an open access article distributed under the terms of the Creative Commons Attribution License, which permits unrestricted use, distribution, reproduction and adaptation in any medium and for any purpose provided that it is properly attributed. For attribution, the original author(s), title, publication source (PeerJ) and either DOI or URL of the article must be cited.
License URL: https://creativecommons.org/licenses/by/4.0/

Keywords: Denitrification, Biofilm, Carbon-to-nitrogen ratio, Marine, Methylotrophs, Methylophaga, Nitrate, Methanol

Funding: Natural Sciences and Engineering Research Council of Canada # RGPIN-2016-06061 This work was supported by the Natural Sciences and Engineering Research Council of Canada: # RGPIN-2016-06061. The funders had no role in study design, data collection and analysis, decision to publish, or preparation of the manuscript.

==============================
Background

Nitrate (NO3−) can accumulate in closed-circuit ecosystems to a toxic level. Adding heterotrophic denitrification process to the water treatment is a strategy to reduce this level. This type of process usually requires the addition of a carbon source. Carbon-to-nitrogen ratio (C/N) is a key parameter known to influence both the function and the activity of microbial communities in bioprocesses. Few studies have examined the influence of C/N on denitrification systems operated under methylotrophic and marine conditions. Here we assessed the influence of C/N (methanol and NO3−) on the performance of a laboratory-scale, recirculating denitrifying reactor operated under marine conditions. We monitored the evolution of the bacterial community in the biofilm to assess its stability during the operating conditions. Finally, the relative gene expression profiles of Methylophaga nitratireducenticrescens strain GP59, the main denitrifier in the denitrifying biofilm, were determined during the operating conditions and compared with those of GP59 planktonic pure cultures.

Methodology

A 500-mL methanol-fed recirculating denitrification reactor operated under marine conditions and colonized by a naturally occurring multispecies denitrifying biofilm was subjected to eight different C/N. We monitored several physico-chemical parameters (denitrifying activities, methanol consumption, CO2 production) throughout the operating conditions. The evolution of the bacterial community in the biofilm during these conditions was determined by 16S rRNA gene amplicon sequencing. Metatranscriptomes were derived from the biofilm to determine (1) the relative gene expression profiles of strain GP59, and (2) the functional diversity of the active microorganisms in the biofilm.

Results

Changes in C/N did not correlate with the denitrification dynamics (NO3− and NO2− reduction rates, NO2− and N2O dynamics), but did correlate with the methanol consumption rates, and the CO2 production rates. Throughout the operating conditions, nitrite and N2O appeared transiently, and ammonium was not observed. The bacterial community in the reactor increased in diversity with biofilm aging, especially among heterotrophic bacteria, at the expense of methylotrophic bacteria. The relative expression profiles of strain GP59 in the biofilm are distinct from those of planktonic pure cultures of strain GP59, and that the expression of several riboswitches and xoxF would be involved in these differences.

Conclusions

When the biofilm community is well established in the reactor, it can withstand changes in C/N with limited impact on the denitrification performance. The increase in the proportion of heterotrophs would allow the reactor to be more flexible regarding carbon sources. This knowledge can be useful for improving the efficiency of denitrification system treating close circuit systems such as marine recirculating aquaculture wastewater or seawater aquarium.

Introduction

Portions of this text were previously published as part of a preprint https://www.biorxiv.org/content/10.1101/2025.05.02.651252v1.full).

Nitrate (NO3−) can pose a threat to aquatic ecosystems (Rouse, Bishop & Struger, 1999; Camargo, Alonso & Salamanca, 2005), as its accumulation can lead to eutrophication of surface waters. This is also true for closed-circuit ecosystems such as aquarium or recirculating aquaculture systems (RAS), where NO3− can reach a high level (>200 ppm) (Grguric & Coston, 1998) and leads to long-term negative effects on the ecosystem (Camargo, Alonso & Salamanca, 2005). In this case, water replacement must be carried out, which is costly and stressful for the ecosystem (Martins et al., 2010; Davidson et al., 2017). Due to the high solubility of NO3−, the main water treatment of these aquatic systems does not efficiently remove NO3−, and a denitrification system has to be added to remediate to the problem (Müller-Belecke et al., 2013).

Denitrification is a microbial process where N oxides serve as terminal electron acceptors instead of oxygen (O2) for energy production when O2 depletion occurs, leading to the production of gaseous nitrogen (N2). Four sequential reactions are required where NO3− is reduced to N2, via nitrite (NO2−), nitric oxide (NO) and nitrous oxide (N2O), and each of these reactions is catalyzed by different enzymes, namely NO3− reductases (Nar and Nap), NO2− reductases (NirS and NirK), NO reductases (Nor) and N2O reductases (Nos) (Philippot & Hojberg, 1999; Richardson et al., 2001, Kraft, Strous & Tegetmeyer, 2011). Different denitrification processes and strategies have been developed over decades, each in response to specific characteristics of infrastructure and water to treat (Ni et al., 2016).

Bioprocesses are composed of an amalgam of microorganisms arranged in a microbial community involved in achieving the biochemical reactions necessary for transformations of the pollutants of interest. The microbial community is in general impacted in the composition of its populations by the operating conditions of the bioprocess, such as, among others, the water type (e.g., freshwater, seawater), the type of molecules to transform, the nutrients provided for electron donors, the carbon source, co-factors, and electron acceptors like O2 under oxic conditions or N oxides (e.g., NO3−) under anoxic conditions. For instances, methanol-fed denitrification processes are composed mostly of methylotrophic bacteria (used C1 carbon as carbon source), which are different from acetate-fed processes, where mainly heterotrophic bacteria composed their microbial communities (Lu, Chandran & Stensel, 2014).

Heterotrophic/methylotrophic denitrification processes (as opposed to autotrophic denitrification) need a readily carbon source to achieve denitrification (Ni et al., 2016; Fu et al., 2022; Brozincevic et al., 2024). As readily available carbon can be low in water system to treat, external carbon has to be added such as acetate, ethanol and methanol. Methanol has the advantage to be not costly and available. The methanol-fed processes produce low biomass; thus, a higher proportion of carbon source is used as electron donors for the denitrification (Ni et al., 2016). However, a methylotrophic community has to establish in these systems, which may pose a problem of lagging time, and are not necessarily flexible as only C1 carbon can be used, contrary to the heterotrophic systems (Sun et al., 2025).

One of the factors that can affect denitrification performance is the appropriate level of carbon to feed the bioprocess for optimal activities. The carbon-to-nitrogen ratio (C/N) is an essential parameter, known to influence both the functions and the activities of microbial communities in bioprocesses (Lu, Chandran & Stensel, 2014). The concentration of the carbon source (methanol was used for this study) has to be properly adjusted according to the level of NO3−entering the denitrification system. The stoichiometry of the denitrification reaction with methanol, 5CH3OH + 6NO3− = >3N2 + 5CO2 +7H2O + 6OH−, requires in theory 0.714 g-C methanol/g-N NO3− (C/N of 0.7) to achieve complete denitrification. However, since denitrification is a microbial process, where methanol and NO3− are also involved in assimilation for cell growth and maintenance, and nitrite and O2 may be present in the process water, a higher level of methanol is required (McCarty, Beck & St. Amant, 1969; Tchobanoglous, Burton & Stensel, 2003).

Inadequate methanol dosage will generate non-metabolized methanol (over dosage) that could pollute the effluent and potentially harm the ecosystem. In addition, suboptimal dosage may result in incomplete denitrification, releasing toxic molecules such as N2O, NO2−, or H2S especially in marine ecosystems (high level of sulfate), or induce the dissimilatory NO3− reduction to ammonium (DNRA) pathway (Her & Huang, 1995; Cattaneo, Nicolella & Rovatti, 2003; Yang, Wang & Zhou , 2012; Mohan et al., 2016). Numerous studies have reported a wide range of optimal C/N with various types of denitrification processes (Fu et al., 2022; Brozincevic et al., 2024). For instances, Sun et al. (2025) studied a lab-scale continuous denitrification reactor inoculated with wastewater sludge and fed with methanol, acetate, glycerol or glucose, at different C/N (3, 6 and 9). They showed that at lower carbon/nitrogen (C/N), accumulation of NO2− occurred. In a methanol-fed denitrification reactor treating a marine RAS (sea bass) operating under continuous conditions, Torno et al. (2018) showed that optimal C/N of 2.1 to 2.3 resulted in 75% total nitrogen removal. However, they found that adjusting the hydraulic retention time for better denitrification performance decreased the denitrification rates. Yin & Guo (2022) and Her & Huang (1995) tested different C/N on methanol-fed sequencing batch reactors (SBR) with acclimated sludge for denitrification. They showed that with C/N of 1.1 and 0.9, respectively, the reactors achieved close to 100% nitrogen removal. Therefore, the proper balance between the contribution of each of these resources is essential for sustainable and cost-efficiency denitrification process.

We have been studying for the last two decades the microbial ecology of a denitrifying biofilm derived from a fluidized-bed, methanol-fed denitrification system operating in continuous mode for treating the seawater basin at the Montreal Biodome natural museum. Autochthonous microorganisms present in the basin that could adapt to the reactor denitrifying environment colonized the surface of the supports. This biofilm is composed of an autochthonous community of at least 15 bacterial species and protozoan (Labbé et al., 2003, Labbé et al., 2007; Laurin et al., 2008), from which several strains were isolated; among others, those involved in denitrifying activities are Hyphomicrobium nitrativorans strain NL23, as well as Methylophaga nitratireducenticrescens strains JAM1 and GP59 (Auclair et al., 2010; Martineau et al., 2013; Villeneuve et al., 2013; Geoffroy et al., 2018). We use this denitrifying biofilm as a model to study how the microbial communities of bioprocesses evolve in response to changes during bioprocess operations. For instance, we cultivated the biofilm at laboratory-scale (vials) to assess its denitrification performance towards physico-chemical changes such as the type of medium, the temperature, pH, and the concentrations of NaCl, methanol and NO3−, and to further assess the impact of these changes on the microbial community (Payette et al., 2019; Villemur et al., 2019). Our results demonstrated that, when the microbial community is well-established, it remains stable and can sustain denitrifying activities despite of all these changes. However, this approach involved static, batch culture assays and artificial seawater, which differed significantly from the continuous operating conditions and the seawater medium of the original denitrification system (Cucaita, Piochon & Villemur, 2021).

Although numerous studies have evaluated the effect of C/N on various types of denitrification systems, relatively few studies have examined the influence of C/N changes on these systems operated under methylotrophic and marine conditions. Investigating this effect can lead to determine optimal conditions of denitrification systems treating closed circuit system such as RAS (Feng et al., 2023) or seawater aquarium like the one that was present at the Montreal Biodome. Here, we established fresh biofilm in a methanol-fed recirculating reactor inoculated with the biofilm taken from the Biodome denitrification system and operated under marine and anoxic conditions. As a first objective, we evaluated the influence of C/N (here methanol and NO3−) on the denitrification performance of the reactor. The reactor was operating sequentially under eight different C/N conditions, during which several physico-chemical parameters were measured. A second objective was to monitor the evolution of the bacterial community in the biofilm in order to assess its stability under the operating conditions. Two approaches were used. First, amplicon sequencing of polymerase chain reaction (PCR)-amplified 16S ribosomal RNA (rRNA) gene sequences were performed to derive the bacterial diversity and to associate bacterial taxa with denitrifying activities. Second, we performed RNA sequencing (metatranscriptome) from biofilm samples to derive (1) the diversity of the active microbial populations, and (2) the functional diversity, which indicated the potential metabolic pathways occurring in microbial community. In a third objective, we took the opportunity of comparing the gene expression profiles of M. nitratireducenticrescens strain GP59 in the recirculating reactor with those obtained in previous works, planktonic cultures for instance (Payette et al., 2019; Villemur et al., 2019; Lestin & Villemur, 2024), to broaden our study to assess the impact of the physiology and the operating modes on these expression profiles. Reads from the metatranscriptomes associated with strain GP59 were retrieved to derive its relative gene expression profiles. A better understanding of how C/N can impact the denitrification performance under methylotrophic, marine conditions, and how the microbial populations response to these changes, will provide useful information for optimizing these denitrification processes.

Material and Methods

Acclimation of denitrifying biomass in recirculating reactor

The medium was the commercial seawater Instant Ocean (IO) medium (Aquarium systems; Mentor, OH, USA), the same used in the original denitrification process. The IO medium (30 g/L) was autoclaved, and one mL/L of sterilized trace element solution (FeSO4•7H2O 0.9 g/L; CuSO4•5H2O 0.03 g/L; MnCl2 •4H2O 0.234 g/L; Na2MoO4•2H2O 0.363 g/L) was added.

The denitrifying biomass was provided in the form of biofilm, immobilized on polyethylene supports of the Bioflow 9-type (1.2 mm diameter) that had developed in the denitrification reactor at the Montreal Biodome, but was preserved in 20% (v/v) glycerol (in IO medium) at −20 °C (Payette et al., 2019). Frozen supports were thawed and submerged for 5 days in the IO medium at 4 °C, supplemented with 0.3% (v/v) methanol. The biofilm was also scraped from the supports and inoculated in the same medium. The biomass free or attached to the supports was transferred to the reactor with seventy empty Bioflow 9-type supports. These empty supports were previously washed in a 35% hydrochloric acid bath overnight and then rinsed a minimum of three times with demineralized water.

The reactor consisted of a glass vessel with an airtight 550-mL working volume, installed in a glove box continuously purged with nitrogen gas and equipped with an oximeter to confirm the absence of O2 (Cucaita, Piochon & Villemur, 2021). A 500-mL volume of culture medium supplemented with 0.3% methanol and 21.4 mM NaNO3 (300 mg-N/L) (C/N of 3) was added to the reactor. A 4-L tank filled with the same medium was appended to the circuit during the acclimation period. The medium was recirculated with a peristaltic pump at 20 mL/min, and the reactor was run at room temperature (ca. 22 °C). The excessive volume of gas produced due to the denitrifying activities was evacuated in a graduated bottle inverted in water and connected to the reactor. Gas production was recorded by water displacement. All equipment, including culture medium, was sterilized before the start of the experiment; however, the operation of the reactor was not carried out under sterile conditions, although it was operated in closed environment of the glove box. The culture medium was replaced regularly as soon as NO3− was exhausted. When biofilm was apparent on the empty supports, supports with the original biofilm were removed. When denitrifying activities (removal of NO3− and NO2−) was stable, the reactor was emptied, and half supports were set aside and stored at 4 °C in IO medium for future used. From this, the C/N assays were carried out with appropriate NO3− and methanol concentrations.

C/N assays

The reactor was run sequentially under eight different C/N ranging from 1.5 to 7.5 according to Table 1, by adjusting the concentration of either NO3− or methanol. The maximum addition of Na+ concentration with NaNO3 in the reaction was 42.8 mM, which added less than 10% Na+ equivalent in the reactor (30 g/L salt, 513 mM NaCl equivalent). The reactor was operated with medium recirculated at 20 mL/min (not with the 4L tank) in sequential batch mode, according to a cycle of three phases: a filling phase, a reaction phase that lasted until complete NO3− reduction, and an emptying phase. The eight conditions were applied twice, which we refer as Period 1 (P1) and Period 2 (P2), with the same biofilm supports (no new inoculation). The reactor ran from November 13, 2019, to June 14, 2020 (31 weeks). During Condition 6 (Period 2), the running operation was abruptly interrupted because of the COVID-19 crisis, which lasted 68 days, during which the supports were transferred at 4 °C in IO medium containing NO3− and methanol. During P1, the operating time for each condition ranged from 96 to 288 h with 1 to 3 medium changes. During P2, the same conditions were reapplied over a shorter interval for Conditions 1 to 5 (48 to 72h) as each condition was running with no medium changes. For Conditions 6 to 8, the reactor was running during much longer time (120 to 240 h) because of the decrease in the denitrification performance probably due to the storing period (COVID crisis).

Table 1 Operating conditions of the reactor.

Conditions	NO3−
(mM)	Methanol
(v/v, %)	C/N
ratio	P1
h	P2
h	
C1	10.7	0.075	1.5	96 (2)	48 (1)	
C2	10.7	0.15	3.0	192 (4)	48 (1)	
C3	21.4	0.15	1.5	144 (2)	72 (1)	
C4	21.4	0.3	3.0	144 (2)	72 (1)	
C5	21.4	0.45	4.5	144 (2)	72 (1)	
C6	21.4	0.75	7.5	144 (2)	240 (2)	
C7	32.1	0.45	3.0	288 (2)	120 (1)	
C8	42.8	0.6	3.0	192 (2)	192 (1)	
Notes.

The reactor was first run sequentially from Condition 1 (C1) to Condition 8 (C8) and referred as Period 1 (P1). The reactor was then operated again under these eight conditions and referred as Period 2 (P2). During P2, Condition 6 (C6-2), the reactor operation was stopped abruptly, and the reactor was set aside at 4 °C for several weeks because of the COVID-19 crisis. After this unfortunate situation, the reactor operation was resumed under Condition 6. The columns P1 and P2 refer to the operating time in hours and number under parentheses is the number of times the reactor was operated with fresh medium.

Liquid samples (one mL) were taken at sampling ports and centrifuged; the supernatant was used to determine the concentrations NO3−, NO2−, ammonium and methanol. The pellets of some of these samples were kept for DNA or protein (or both) extractions. A 10-mL volume of gas was also taken from the head space to measure N2O and CO2.

At the end of each condition at the time of the emptying phase, three supports were taken to determine the protein content of the biofilm and to extract the nucleic acids (DNA and RNA). These supports were replaced by three others from the supports that were set aside previously, in order to keep a fixed number of supports. At the emptying phase, the medium was also recovered to collect the suspended biomass and determine its protein content. The biomass collected to extract the RNAs was scraped from supports under the anoxic environment of the glove box and immersed in the extraction buffer (50 mM Tris-HCl, 100 mM EDTA, 150 mM NaCl, pH 8.0) and water-saturated phenol pH 4.3 (v/v). The sample tubes were then immediately frozen in liquid nitrogen and stored at −70 °C until use. For DNA extraction and total protein assays, the biomass from the supports was scraped and then frozen at −20 °C in five mL of sterile culture medium until use. To recover the biomass in suspension, 300 mL of the culture medium was centrifuged at 16,000× g and the pellet was mixed with five mL of sterile medium and stored at −20 °C until use.

Analytic methods

Measurements of the concentrations of NO3−, NO2− and protein content, and the determination of the specific NO3− reduction rates and the specific NOx (NO3− + NO2−) rates were carried out according to Cucaita, Piochon & Villemur (2021). N2O and CO2 concentrations (in ppmv) in the headspace were determined by gas chromatography. Headspace samples (10 mL) were collected using a Pressure Lok gastight glass syringe (VICI Precision Sampling Inc., Baton Rouge, LA, USA) and were injected through the injection port of a gas chromatograph equipped with a thermal conductivity detector and electron-capture detector (7890B seriesGCCustom, SP1 option 7890-0504/0537; Agilent Technologies). The amount of N2O (in µmole) in the reactor was calculated as described in Mauffrey et al. (2017). The NO2− and N2O showed transitory dynamics with appearance that peaked after a certain time, then reduced completely (Figs. 1G and 1H). To quantify these dynamics, we calculated the area under the curve of the amount of NO2− or N2O by the time (mM-h) they appeared and then disappeared (Data S1). The CO2 production rates were calculated from the linear portion of CO2 produced by the time, which occurred mainly during the first 5–15 h (Data S1). This period corresponded when NO3− reduction was occurring.

Figure 1 Physico-chemical measurements during the different operating conditions.

(A–F) Physico-chemical parameters measured during the operating conditions (C1 to C8) for both periods (P1 and P2). (G & H) Time-course profiles of NOx, NO3−, NO2− and N2O. Dot lines: Log normal bestfit and Gaussian bestfit for N2O and NO2−measurements, respectively, used to calculate the aera under the curve for the N2O and NO2− dynamics. Residuals: % of NOx, NO3− and NO2− from the initial methanol concentrations (i-NO3−) that remained in the reactor after four operating days. Methanol consumption rates followed first order dynamics. Methanol consumed: % of methanol consumed at the end of the period. nm: not measured. See Table 1 for conditions’ nomenclature.

Methanol was measured by gas chromatography coupled with a flame ionization detector (GC-FID). Nitrogen gas was used as a carrier gas. The culture medium samples were supplemented with 5% isopropanol as an internal standard. The standard range included solutions of 0.01 to 0.2% methanol. The total amount of methanol consumed was determined at the end of the reaction phase. The methanol consumption rates followed a first order dynamics, where a k factor was calculated as [ln(Ct/C0)]/time with Ct methanol concentration at the sampling time and C0 the initial methanol concentration (Data S1). Ammonium concentration was measured by colorimetric method as described by Mulvaney (1996).

Polymerase chain reaction-denaturing gradient gel electrophoresis and 16S rRNA amplicon sequencing

Supports were taken at the end of each condition and period for a total of 16 samples. Biofilm from the original biofilm was also taken for DNA extraction. DNA extractions of biomass on supports were performed as described (Geoffroy et al., 2018). The PCR amplifications (one PCR for each sample) of the V3 region of the 16S rRNA genes for denaturing gradient gel electrophoresis (DGGE) experiments were performed as described (Lafortune et al., 2009) with the 341f and 534r primers.

The bacterial composition of the biofilm was determined by amplicon sequencing of the 16S rRNA V3–V4 regions with the S-D-Bact-0341-b-S-17 (5′TCGTCGGCAGCGTCAGAT GTGTATAAGAGACAG-CCTACGGGNGGCWGCAG 3′) and S-D-Bact-0785-a-A-21 (GTCTCGTGGGCTCGGAGATGTGTATAAGAGACAG-GACTACHVGGGTATCTAAT CC) primers (Klindworth et al., 2013) linked with Illumina sequences (underlined sequences). The amplifications were carried out in 25-µL volume containing the commercial Taq polymerase buffer, 0.25 mM MgSO4, 0.2 µg BSA, 200 nM of each primer, 0.5-unit AccuPrime™ Taq DNA Polymerase (Invitrogen, Carlsbad, CA, USA) and 50 ng DNA extract. The reactions were run at 94 °C 5 min, 30 cycles at 94 °C 30 s, 55 °C 30 s, and 68 °C 30 s, and finally at 68 °C 10 min. Residual primers and primer dimers were removed with the AMPure XP magnetic beads kit according to the manufacturer (Beckman Coulter, USA). The amplicons were then amplified a second time in which indexes were added. The amplifications were carried out in a 25-µL volume containing the Taq polymerase buffer, 0.25 mM MgSO4, 0.2 µg BSA, 400 nM index primers, 0.5-unit AccuPrime™ Taq DNA Polymerase (Invitrogen, Carlsbad, CA, USA) and 5 µL amplicon. The reactions were run at 94 °C 3 min, 8 cycles at 94 °C 30 s, 55 °C 30 s, and 68 °C 30 s, and finally at 68 °C 5 min. Residual primers and primer dimers were once again removed with AMPure XP magnetic beads kit, and the purified amplicons were quantified with the Quant-iT PicoGreen dsDNA Assay kit (Invitrogen). The amplicons were sent for sequencing by Illumina MiSeq PE250 (250 bp paired-end sequencing reactions) at the Centre d’expertise et de services Génome Québec (Montréal, QC, Canada). Sequence quality per base was >32. Sequencing reads were processed by the dada2 pipeline at the Galaxy server (https://usegalaxy.eu/) to generate ASVs (amplicon sequence variants). The parameters were: (1) FilterAndTrim of fastqR1 and R2 with Truncate reads at quality threshold set to 2, Trim start of each read set to 15, and Remove short reads set to 20; (2) learn errors with Magnitude of number of bases to use for learning set to 8, Error function set to loess to estimate error rates from transition counts, and Maximum number of times to step through the selfconsistency loop set to 10; (3) dada: Remove sequencing errors, (4) mergePairs with Minimum length of the overlap set to 12; (5) removeBimeraDenovo, with homology set to 80%. The taxonomic assignment was carried out at the Silva web site (https://www.arb-silva.de/) with sequence identity cut off set to 80%. Number of reads ranged from 27 268 to 33 031 for the recirculating reactor samples, and 423 208 reads from the original biofilm sample. Rarefaction analyses were performed with the Analytic Rarefaction v1.3 software downloaded from the UGA Stratigraphy Lab web site (http://stratigrafia.org/) with increments of 10. All samples showed rarefaction saturation. ASVs affiliated to the same lineage were grouped together, in order to establish the proportions of each taxon in the samples. Raw sequencing data were deposited in Sequence Read Archive (SRA) at the National Center for Biotechnology Information (NCBI: https://www.ncbi.nlm.nih.gov/) (Data S2).

Metatranscriptomes

The RNA extraction was previously described by Mauffrey, Martineau & Villemur (2015). RNA quality was verified by agarose gel electrophoresis and by spectrophotometry (Nanodrop) with ratio OD 260/OD 280 > 1.8. The RNA samples were sent for sequencing using the Illumina Method (NovaSeq 6000 S4 PE100). Library preparation and sequencing were performed by the Centre d’expertise et de services Génome Québec (Montréal, QC, Canada). Ribosomal RNA were depleted using the Ribo-ZeroTM rRNA Removal Kits (Meta-Bacteria; Epicentre, Madison, WI, USA). RNAseq reads were deposited in SRA (Data S2). Raw sequencing reads were trimmed using fastq_quality_filter (1.0.2) to remove low quality reads (score < 20) (Gordon & Hannon, 2010) , and the paired reads were then merged.

Relative gene expression profiles of strain GP59

The paired reads were aligned to the genome and plasmids of M. nitratireducenticrescens strain GP59 (GenBank accession number CP021973.1, CP021974.1, CP021975.1) using Bowtie2 (v 2.5.0) (Langmead & Salzberg, 2012) and annotated with Bedtools (v 2.30.0) (Quinlan & Hall, 2010). Genes that were significantly differentially expressed were identified by EdgeR (v 3.36.0) (Robinson, McCarthy & Smyth, 2010) with trimmed mean of M values (TMM) method to normalize library sizes (robust = TRUE; P-value adjusted threshold = 0.05; P-value adjusted method = Benjamini and Hochberg). All these analyses were performed on the Galaxy server (https://usegalaxy.org/). Genes were considered differentially expressed when the false discovery rate (FDR) was ≤ 0.05.

Analysis of the unaligned RNA reads

The paired reads were aligned using Bowtie2 to a concatenated sequence consisting of the three reference genomes (M. nitratireducenticrescens strain JAM1 (GenBank accession number CP003390.3) + strain GP59 + H. nitrativorans strain NL23 (CP006912)) and the two GP59 plasmids. The reads that did not align were de novo assembled by Trinity v. 2.4.0 (Grabherr, Haas & Yassour, 2011), with Strand specific data set to No, In silico normalization of reads set to Yes with max. read coverage set to 50, Minimum Contig Length set to 200, Minimum count for K-mers to be assembled set to 1 at the Galaxy server. Estimation of the transcript abundance of the de novo assembled sequences, here named contigs, was performed by RSEM (Galaxy server) (Li & Dewey, 2011), which provided number of transcript reads per contig. Normalization was performed by adding the number of transcript reads by RSEM of all contigs, which was then expressed as by transcripts per million (TPM-RSEM) resulting of relative transcript abundance of the contigs. The resulting contigs were annotated at the Joint Genomic Institute (https://img.jgi.doe.gov/cgi-bin/m/main.cgi) with the IMG Annotation Pipeline v.5.0.25 with Reference Databases: Cath-Funfam v4.1.0/v4.2.0, COG v2014, IMG-NR v20190607, KEGG v77.1, Pfam v30, SMART v01_06_2016, SuperFamily v1.75, TIGRFAMs v15.0) to find open reading frames (ORFs) with their putative function and affiliation (Data S2).

Contigs containing ORF (contig-ORF) and showing affiliation to a microbial lineage were grouped by taxon and their TPM-RSEM were summed. The proportion of each taxon per biofilm sample was then derived. For each identified taxon, the KO enzyme number from deduced function of gene products were retrieved and used to do reconstruction metabolic pathway by KEGG mapper-reconstruct analysis (https://www.genome.jp/kegg/mapper/reconstruct.html). If the identified taxa were also represented by a corresponding species in KEGG database, the metabolic pathways were examined. The carbon assimilation and dissimilation pathways were derived from the KEGG (https://www.genome.jp/kegg/pathway.html) and MetaCyc (https://metacyc.org) databases.

Statistical analysis

Pearson correlation analyses were performed by GraphPad Prism v10.4.2. Relative gene expression profiles of strain GP59 were analyzed by Principal coordinate analysis (PCoA) (Canoco version 5.15; Ter Braak & Šmilauer, 2018), with Log (1 * X + 1) calculating matrix of distances, using percentage difference (Bray–Curtis distance). PERMANOVA (Vegan package version 2.6.2, command Adonis2) were performed for significance with 999 permutations (Anderson, 2017).

Results

Acclimation of the denitrifying biomass

The Biodome denitrification system was dismantle in 2006, from which we took all the biofilm supports and preserved them in glycerol at −20 °C. Previous works by Laurin et al. (2006) demonstrated no differences in the recovery of the denitrifying activities of the biofilm when it was stored at −20 °C or −70 °C for 17 months. Thirteen year later, the frozen supports were used to inoculate the recirculating reactor for the colonization of new supports. This strategy has been successfully used in our previous studies (Payette et al., 2019; Villemur et al., 2019). We refer to this biofilm as the Original Biofilm to distinguish it from the one that developed in the recirculating reactor. The reactor was operated under sequential-batch conditions with the IO medium, the same as that used by the Biodome, and supplemented with 0.3% methanol and 21.4 mM NO3− (0.891 g-C/L and 0.2996 g-N/L: C/N of 3). These concentrations were based on a previous report by Payette et al. (2019) that succeeded culturing the biofilm under these conditions. After 37 days, stable denitrifying activities (reduction of NO3− and NO2−) occurred in the reactor, and fresh biofilm was visually observed on the supports (Fig. S1). This biofilm is referred to as the RR biofilm (RR for recirculating reactor). The C/N assays were then carried out. Day 0 (D0) was set after the 37-day acclimation period.

C/N assays

The reactor with supports fully colonized by denitrifying biomass was operated sequentially under eight different C/N (1.5 to 7.5) (Table 1). These conditions were performed twice (P1 and P2) with the same biomass to assess the reproducibility of the results. The reactor was operated for 31 weeks during these two periods, where several physico-chemical parameters were measured (Figs. 1A to 1H). The pH was stable (7.6–8.0) and the protein content on the supports (35), representing the total microbial biomass, ranged from 10 to 25 mg. No ammonium was detected under any conditions.

NO3− reduction was occurring with transient NO2− and N2O accumulation (Figs. 1C and 1D and 1H; Data S1). On average, the NOx (NO3− + NO2−) reduction rates (Fig. 1A) were 77% of the NO3− reduction rates (Fig. 1B), due to the transient NO2− accumulation (Fig. 1C). Although we operated the reactor twice under all eight C/N ratios, we encountered a significant obstacle, the COVID-19 crisis, where the reactor had to be abruptly shut down during P2-Condition 6, and stored in cold room for several weeks, which negatively impacted the denitrification performance (Fig. 1). If we assume the NOx reduction rates represent the denitrification rates (NO and N2O reduction have minimal influence on these rates), the performance of the reactor ranged from 0.166 to 0.328 mM h−1 if we exclude post-COVID data. These rates averaged 0.119 mM h−1 in P2-Conditions 6 to 8. We estimated the residuals of NOx, NO3− and NO2− after four operating days (Figs. 1A–1C). We observed these residuals in P1 only under Conditions 7 and 8, with 28.5% residual NOx in Condition 8. In P2, Conditions 6 to 8 showed higher levels of residuals due to the lower reactor performance with 50.6% residual NOx in Condition 8.

The peak of N2O concentrations appeared mainly in the first 10-20 h of operating conditions and ranged from 0.04 mM to 0.4 mM (Fig. 1G and 1H). No residual N2O was found at the end of each condition. The amount of methanol consumed ranged from 0.031% to 0.147% (v/v), which corresponded to between 9% to 53% of the initial methanol concentrations (i-Methanol) (Fig. 1E). The methanol consumption rates (Fig. 1E) followed first-order dynamics, while the CO2 production rates (Fig. 1F) were at their maximum during the first 5-15 h (Data S1).

Correlation analyses were first performed between (i) the i-Methanol, (ii) the initial NO3− concentrations (i-NO3−), (iii) the C/N applied to the reactor, and the physico-chemical parameters measured during the operating conditions (Table 2). These analyses revealed that changes in C/N did not correlate with the denitrification dynamics (NO3− reduction rates, the NOx reduction rates, the NO2− and N2O dynamics; p > 0.05). However, these C/N changes did correlate with the methanol consumption rates, and the CO2 production rates (0.05 < p < 0.001; Table 2). Correlations (p < 0.05) occurred between the i-NO3− and (i) the NO3− reduction rates, (ii) the NOx reduction rates and (iii) the NO2−dynamics (Table 2), meaning higher i-NO3− resulted in higher denitrification dynamics. The i-Methanol correlated with the methanol consumption rates and the CO2 production rates (0.05 < p <0.001; Table 2), meaning higher i-Methanol resulted in higher carbon dynamics. Correlation analyses were also performed between the respective measured parameters (Table 2). As expected, strong correlation (p < 0.0001) was observed between the NO3− reduction rates and the NOx reduction rates (Table 2). The N2O dynamics did not correlate with any conditions.

Table 2 Pearson correlation coefficients between measured parameters.

	NO3− reduction rates	NOx reduction rates	NO2− dynamics	N2O dynamics	Methanol consumption rates	CO2 production rates	
i-NO3−	0.0312	0.0009	<0.0001	ns	ns	ns	
i-Methanol	ns	ns	ns	ns	0.0154	0.0065	
C/N	ns	ns	ns	ns	0.031	0.0015	
NO3− reduction rates		<0.0001	ns	ns	ns	ns	
NOx reduction rates			ns	ns	ns	ns	
NO2− dynamics				ns	ns	ns	
N2O dynamics					ns	ns	
Methanol consumption rates						ns	
Notes.

ns, nonsignificant.

All correlations are directly proportional.

i-Methanol and i-NO3−: Initial methanol and initial NO3− concentrations, respectively.

Data obtained after COVID-19 (C6, C7 and C8, Period 2) were not included in the analysis.

Bacterial community

The evolution of the bacterial community of the RR biofilm during the different operating conditions and the Original Biofilm was assessed by performing PCR amplification of the 16S rRNA genes and by analysing the composition of the amplicons by DGGE and amplicon sequencing. We first compared the DGGE migration profiles of the Original Biofilm between a fresh sample collected in 2004, when the Biodome system was operational, and the frozen stock sample used to inoculate the reactor. The results showed that the two main DNA bands of M. nitratireducenticrescens and H. nitrativorans were present, although the M. nitratireducenticrescens band is more intense (higher proportion) in the 2004 sample. These results suggest that no major changes in the bacterial community occurred in the frozen stock.

The polymerase chain reaction-denaturing gradient gel electrophoresis (PCR-DGGE) migration profiles showed striking differences between the Original Biofilm sample and the samples collected from the reactor (Fig. 2A). The same migration profiles were obtained with samples collected during the second period, suggesting that the interruption of the reactor during the COVID crisis did not deeply affect the composition of the bacterial community. The migration profile of the D0 sample revealed some similarities with those derived from samples collected during the operating conditions. The PCR-DGGE migration profiles of the later samples did not reveal substantial changes between them, suggesting that the bacterial community remained stable during the operating conditions.

Figure 2 Bacterial profiles in the biofilm samples.

(A) DNA was extracted from biofilm samples taken from the reactor at Day 0 (D0) and during Period 1, conditions 1 to 8 (C1 to C8), and taken from the Original Biofilm (OB). The V3 region of the 16S rRNA genes were amplified with one primer having a GC clip, and the resulting amplicons were resolved by denaturing gradient gel electrophoresis. Mn and Hn: Locations of the migration of the V3 sequences of Methylophaga nitratireducenticrescens and Hyphomicrobium nitrativorans, respectively. The 2004 lane is PCR-DDGE made in February 2004 on DNA extracted from the biofilm of the Biodome denitrification system when it was operational. (B) DNA were extracted from the recirculating reactor at D0 and during the operating conditions: Condition 5 (P1 and P2; C5-1, C5-2) and Condition 6 (P2; C6-2). For comparison, we added data from the Original Biofilm (OB). Part of the 16S rRNA gene was amplified and the resulting amplicons were sequenced. Taxonomic affiliations of the resulting sequences were carried out, and their proportions derived. The most relative abundant taxa are illustrated. (C) Principal coordinate analysis (PCoA) of the bacterial profiles of the five biofilm samples was carried out using Bray–Curtis distance calculation Muricauda was reclassified as Flagellimonas.

We performed 16S amplicon sequencing on total DNA extracted from the RR biofilm samples. The first sample was collected at D0 (C/N of 3.0; 21.4 mM NO3−, 0.3% methanol) to provide the bacterial diversity profile at day 0. Two samples came from the same conditions (C-5, 21.4 mM NO3− and 0.45% methanol: C/N of 4.5) but one in P1 (C5-1; Day 30) and the second in P2 (C5-2; Day 120). Finally, the sample from condition 6 (21.4 mM NO3− and 0.75% methanol; C/N of 7.5) during P2 (C6-2; Day 200) was chosen because it came from the reactor operated after the COVID break. For comparison, we added data that we obtained from the Original Biofilm of the Biodome denitrification system (estimated C/N of 0.86) (Labbé, Parent & Villemur, 2003).

We identified forty-two taxa among the five biofilm samples (Data S3) and determined their proportions in the biofilm samples. Figure 2B illustrates the proportions of the most abundant taxa observed in the samples. Principal coordinate analysis (PCoA; Data S3) was performed on the bacterial population profiles of these samples. Due to the uniqueness of the samples (no replicate), we could not perform statistical analysis such as PERMANOVA to determine the significance of variations between the five profiles. However, we believe it is reasonable to conclude that the bacterial population profiles of the C5-1, C5-2 and C6-2 biofilm samples are similar, and those of the D0 and Original Biofilm samples differ from each other, and from the other ones (Fig. 2C). These results concur with the PCR-DGGE results. Therefore, for the analysis of the composition of the bacterial community in the biofilm samples, we considered the C5-1, C5-2 and C6-2 bacterial profiles as a whole, distinct from those of the Original Biofilm sample and the D0 biofilm sample.

Thus, higher proportions of Hyphomicrobiales, Roseovarius spp. and Oceanibaculaceae, and a much lower proportion of Methylophaga spp. were found in the Original Biofilm sample compared to the four RR biofilm samples. In the D0 biofilm sample, the highest relative abundance of Methylophaga spp. (69%) was found, which is consistent with the DGGE migration profile of the D0 biofilm sample, where an intense DNA fragment associated with M. nitratireducenticrescens was observed. For the C5-1, C5-2 and C6-2 biofilm samples, increases in the relative abundance of Marinicella spp., Stappiaceae spp., Muricauda (Flagellimonas) spp. and Oceanibaculaceae were observed compared to the D0 biofilm sample with an approximately 50% decrease in relative abundance of Methylophaga spp. These three biofilm samples had higher alpha diversity indexes compared to the Original Biofilm (Table 3; Data S3). Due to the high proportion of Methylophaga spp. in the D0 biofilm sample, the alpha diversity indexes were lower than those of the four other biofilm samples. Regarding beta diversity (Table 3), examination of common taxa between samples revealed that taxa found in the RR biofilm samples shared the highest number of taxa. All these results showed that the original bacterial community evolved through diversification in the recirculating reactor from Day 0, and then remained relatively stable during the operating conditions.

Table 3 Diversity indexes of taxa deduced from 16S amplicon sequencing.

Alpha diversity indexes	
	OB	D0	C5-1	C5-2	C6-2	
Richness	27	33	39	36	37	
Shannon	1.82	1.43	2.35	2.39	2.14	
Simpson (1-H)	0.752	0.516	0.852	0.848	0.803	
Pielou	0.553	0.408	0.640	0.666	0.592	
Beta diversity (Jaccard index)	
	OB	D0	C5-1	C5-2	C6-2	
OB	1.000	0.500	0.610	0.575	0.600	
D0		1.000	0.756	0.683	0.707	
C5-1			1.000	0.875	0.900	
C5-2				1.000	0.872	
C6-2					1.000	
Notes.

Richness: number of taxa found in each sample.

Shannon, Simpson, Pielou: higher the values, higher the diversity.

Beta diversity: higher the values, closer the diversity between samples.

OB: Original Biofilm.

Metatranscriptome analysis

We derived metatranscriptomes of the microbial community from four RR biofilm samples to determine (1) the relative transcript profiles of M. nitratireducenticrescens strain GP59, the main denitrifier in the RR biofilm; and (2) the potential metabolic pathways active in the RR biofilm.

The metatranscriptomes were derived during P1 from: Condition 1 (C1-1; C/N of 1.5; 10.7 mM NO3−, 0.075% methanol; Day 4), Condition 4 (C4-1; C/N of 3.0; 21.4 mM NO3−, 0.3% methanol; Day 24). Condition 6 (C6-1; C/N of 7.5; 21.4 mM NO3−, 0.75% methanol; Day 36), and Condition 8 (C8-1; C/N of 3; 42.8 mM NO3−, 0.6% methanol; Day 89). Condition 1 and Condition 6 were chosen as the lowest (1.5) and highest (7.5) C/N applied to the reactor, respectively. Although Condition 4 and Condition 8 had the same C/N (3.0), they were adjusted with distinct levels of NO3− and methanol. In addition, we derived the metatranscriptome of the Original Biofilm for comparison.

Relative transcript profiles of M. nitratireducenticrescens strain GP59

The proportions of transcript reads associated with strain GP59 in the metatranscriptomes were similar across the four RR biofilm RNA samples (average 7.15%), four times higher than in the Original Biofilm RNA sample (Data S4). The relative gene expression profiles of strain GP59 were then derived from the five RNA samples. Due to the uniqueness of the sample (no replicate), we were unable to perform statistical analysis such as PERMANOVA to determine the significance in the variations between the five expression profiles. However, we took this opportunity to compare these profiles with those derived from our previous studies. These include:

1-The Original Biofilm cultured in vials under batch-static conditions in artificial seawater (ASW, 2.75% NaCl), with two different NO3− concentrations and temperatures: < <300N23C> > (21.4 mM NO3−, 0.15% methanol, C/N of 1.5, 23 °C); and < <900N30C> > (64.2 mM NO3−, 0.45% methanol, C/N of 1.5, 30 °C) (Payette et al., 2019; Villemur et al., 2019).

2- The Original Biofilm cultured in vials under batch-static conditions in low salt concentration ASW: < <0%NaCl> > (ASW with 0% NaCl, 21.4 mM NO3−, 0.15% methanol, C/N of 1.5, 23 °C) (Payette et al., 2019; Villemur et al., 2019).

3-Planktonic cultures (triplicate) of strain GP59 cultured in a Methylophaga-specific culture medium (Methylophaga-1403, 0.3% methanol; C/N of 3) under oxic conditions with NO3− (21.4 mM, PlkON) or without NO3− (PlkO), and under anoxic conditions with 21.4 mM NO3− (PlkAN) (Lestin & Villemur, 2024).

All these cultures differed in culture media (IO, ASW, low salt ASW, Methylophaga-1403), C/N, operating mode (batch-static, continuous, recirculating sequential-batch), and physiology (biofilm, planktonic). The objective of this analysis was to reveal metabolism pathways of strain GP59 affected by the different culture conditions. PCoA clearly showed distinct clustering of each category of cultures, explained by 68.5% variations (Fig. 3; Data S5). PERMANOVA were performed with the planktonic cultures (triplicate: Methylophaga-1403 medium, anoxic or oxic, presence or absence of NO3−, planktonic), and the four RR biofilm samples assuming that their relative transcript profiles of the RR biofilm RNA samples were similar enough to be considered as replicates (IO medium, anoxic, biofilm, presence of NO3−). The results showed, as perceived in Fig. 3, that the medium/physiology (IO (biofilm) vs M1403 (planktonic)), the operating conditions (oxic vs anoxic), and the methanol and NO3− concentrations had substantial effects (p < 0.01) on the relative transcript profiles of strain GP59 (Data S5).

Figure 3 Principal coordinate analysis of the relative transcript profiles of strain GP59 in different cultures.

Metatranscriptomes of the four RR biofilm samples, and the Original Biofilm were performed, and the relative transcript profiles of the 3,204 genes of strain GP59 were derived. The strain GP59 gene expression profiles from our previous studies were also included in PCoA. These profiles were (1) the Original Biofilm cultured in ASW medium under different conditions (300N23C; 900N30C, 0%NaCl), and pure cultures of strain GP59 cultured in Methylophaga 1403 medium under oxic or anoxic conditions (PlkO, PlkON, PlkAN). See the text for complete description. PCoA by percentage difference (Bray–Curtis distance) were performed with the Canoco software for ordination version 5.15. Permanova (999 permutations) was performed with profiles of the four RR biofilm samples and the planktonic culture replicates.

We examined the relative transcript levels of genes involved in the N cycle. Thirty-three genes or gene clusters were selected for their involvement in denitrification, N assimilation, regulation and NO-responses (Data S5). PCoA was performed to derive the preferential expression profiles in the different culture types. Figure 4 illustrates the PCoA that explained 78% of the variations. These genes and cultures fell into three specific quadrants: (i) the lower left quadrant with the planktonic oxic cultures (PlkO, PlkON); (ii) the lower right quadrant with the 0%NaCl biofilm (non-marine conditions) and the Original Biofilm; and (iii) upper quadrants with the planktonic anoxic cultures (PlkAN) and the RR biofilm samples. PERMANOVA showed this time that only the media/physiology (IO (biofilm) vs M1403 (planktonic)), and the operating conditions (oxic vs anoxic) had substantial effects (p < 0.01) on the relative transcript profiles of these genes.

Figure 4 Principal coordinate analysis of the relative transcript profiles of genes involved in the N cycle of strain GP59 in different cultures.

Thirty-three genes or gene clusters were identified in strain GP59 genome to be involved in denitrification, N assimilation, regulation and NO-responses. Their relative transcript levels derived from the biofilm metatranscriptomes, or from pure culture transcriptomes (see Fig. 3 legend) was used in PCoA. PCoA by percentage difference (Bray–Curtis distance) were performed with the Canoco software for ordination version 5.15. narXL: NO3− /NO2− transcription regulator; nar1: NO3− reductase 1 operon with two narK transporters; narK12f: NO3− transporter; nar2: NO3− reductase 2 operon; nirK: NO-forming NO2− reductase; nor1: nitric oxide reductase 1 gene cluster; norRE: nitric oxide reductase transcription regulator; nor2: nitric oxide reductase 2 gene cluster; nos: nitrous oxide reductase gene cluster; nasAnirDB: assimilatory NO3−/NO2− reductases; cAMP CRP1,2,3,4: represent four genes encoding Crp/Fnr family transcriptional regulators; fnr: fumarate/ NO3− reduction transcriptional regulator Fnr; nnrS1,2,3: represent three genes encoding protein involved in response to NO; nsrR1,2: represent two genes encoding nitric oxide-sensitive transcriptional repressors; dnrN/ytfE: iron-sulfur cluster repair protein; hmp1,2: represent two genes encoding nitric oxide dioxygenases; ntrXY: Nitrogen regulation protein; AmTrp1,2: represent two genes encoding ammonium transporters; gltBD: glutamate synthase; gdhA: glutamate dehydrogenase; glnE: (glutamate–ammonia-ligase) adenylyltransferase; glnB: P-II family nitrogen regulator; glnK: P-II family nitrogen regulator; glnD: [protein-PII] uridylyltransferase; glnA: glutamate–ammonia ligase. Permanova (999 permutations) was performed with profiles of the four RR biofilm samples and the planktonic culture replicates.

Higher relative transcript levels of several genes in the N assimilation pathways (ammonium transporters, nasAnirDB, glnA, glnK, glnLG) and the denitrification genes nor2 and narK12f were found in the Original Biofilm and the 0%NaCl biofilm (Fig. 4). The regulatory genes, fnr, ntrXY and narXL, and the denitrification genes nos, nirK and nar2 showed higher relative transcript levels in the PlkO and PlkON (oxic) cultures. The PCoA pattern suggests that the relative expression profiles are similar in the PlkAN (anoxic planktonic) cultures and the RR biofilm cultures. For instances, the denitrification genes nar1, nor1 and its associated regulatory genes norRE had higher relation transcript levels in these cultures, as well as nnrS1, nsrR1 and dnrN, which are adjacent genes in strain GP59 genome, and hmp1 and nnrS2 that encode putative regulators or proteins with NO-response function.

We then examined the relative expression profiles of genes involved in the C cycle. Among the 112 genes or gene clusters identified, we selected 17 genes or gene clusters for PCoA, in which their relative transcript levels varied more than five times (significant differences inferred by EdgeR analysis with FDR <0.05) between at least two conditions (Data S5). We also included in the PCoA the relative transcript profiles of the eleven riboswitches (Ribo1 to Ribo11) identified in the genome of strain GP59 (Table 4; Data S7). Ribo1 to Ribo7 are c-di-GMP riboswitches; Ribo9 and Ribo10, cobalamin riboswitches adjacent to each other and flanked by genes involved in cobalamin synthesis; Ribo8 and Ribo11, thiamine pyrophosphate and S-adenosyl methionine riboswitches, respectively. PERMANOVA showed that the media/physiology (IO (biofilm) vs M1403 (planktonic)), the operating conditions (oxic vs anoxic), and the methanol concentrations had substantial effects (p < 0.01) on the relative transcript profiles of these genes.

Table 4 List of riboswitches and their flanked gene.

Riboswitch number and type	Location	Gene product	
Riboswitch1: c-di-GMP	82867	Thrombospondin	
Riboswitch2: c-di-GMP	822774	Pyrroloquinoline quinone precursor peptide PqqA	
Riboswitch3: c-di-GMP	986642	TonB-dependent receptor	
Riboswitch4 c-di-GMP	1390371	VPLPA-CTERM sorting domain	
Riboswitch5 c-di-GMP	1405201	PEP-CTERM domain	
Riboswitch6 c-di-GMP	1406231	PEP-CTERM domain	
Riboswitch7 c-di-GMP	1407172	PEP-CTERM domain	
Riboswitch8: Thiamine pyrophosphate	2153463	Phosphomethylpyrimidine synthase ThiC	
Riboswitch9 and 10: Cobalamin	2296147	Cobalamin synthesis gene cluster	
Riboswitch11: s-adenosylmethionine	3207836	Methionine adenosyltransferase	
Notes.

Riboswitch9 and 10 are adjacent to each other. Location: from genome annotation: GenBank accession number CP021973.1. See Data S7 for complete detail.

The relative transcript levels of the four xoxF genes (encoding putative lanthanide [Ln3+]-dependent methanol dehydrogenases) were higher in all four RR biofilm samples (Fig. 5), as well as two genes encoding the pyruvate carboxylase (pycAB; HCO3− to oxaloacetate) and the carbonic anhydrases (cynT; dissolved CO2 to HCO3−), involved in CO2 assimilation. The relative transcript levels of the methanol dehydrogenase gene cluster (mxa) and one of the formate dehydrogenase gene clusters (fdh2) were higher in the PlkO, PlkON (oxic) cultures. Interestingly, the relative transcript levels of 10 out of 11 riboswitches were higher in the RR biofilm samples (Fig. 5). The other riboswitch (Ribo5; upstream of an ORF encoding PEP-CTERM domain, Table 4) had higher relative transcript levels in the PlkAN (planktonic) cultures.

Figure 5 Principal coordinate analysis of the relative transcript profiles of genes involved in the Carbon cycle and riboswitches of strain GP59 in different cultures.

One hundred twelve genes or gene clusters were identified in strain GP59 genome in the C cycle. Among them, seventeen in which their relative transcript levels varied more than five times between at least two conditions were chosen for PCoA. The relative transcript profiles of the eleven riboswitches (Ribo1 to Ribo11) were also included in PCoA. PCoA by percentage difference (Bray–Curtis distance) were performed with the Canoco software for ordination version 5.15. mxa: methanol dehydrogenase; xoxF1,2,3,4: represent four genes encoding putative lanthanide [Ln 3+]-dependent methanol dehydrogenases ; fae1,2,3: represent three genes encoding 5,6,7,8-tetrahydromethanopterin hydro-lyases; ftr: formylmethanofuran–tetrahydromethanopterin N-formyltransferase; fdh2: formate dehydrogenase; tkt: transketolase; prsA: ribose-phosphate pyrophosphokinase; rbcLS: ribulose-bisphosphate carboxylase; fbp1,2,3: represent three genes encoding fructose-1,6-bisphosphatases; fbaA1,2,3: represent three genes encoding fructose-bisphosphate aldolases; gapA1,2: represent two genes encoding glyceraldehyde 3-phosphate dehydrogenases; pycAB: pyruvate carboxylase; mqo: malate dehydrogenase (quinone); cynT1,2,3: represent three genes encoding carbonic anhydrases; Ribo1 to Ribo7: c-di-GMP riboswitches, Ribo8: thiamine pyrophosphate riboswitch, Ribo9 and 10: cobalamin riboswitches, Ribo11: S-adenosyl methionine riboswitch. Permanova (999 permutations) was performed with profiles of the four RR biofilm samples and the planktonic culture replicates.

Diversity of transcriptionally active bacteria

RNA transcripts from the biofilm are presumed to originate largely from active microorganisms. Metatranscriptome analyses determined the most likely affiliation of the gene products encoded by the assembled sequences (contig), and thus the diversity of the active microorganisms. These data, together with those obtained from 16S amplicon sequencing, provided a more complete picture of the diversity of the biofilm samples.

We analysed reads that were not associated with M. nitratireducenticrescens and Hyphomicrobium nitrativorans to identify the most transcriptionally active taxa in the five biofilm samples, and to assess the potential metabolic pathways used by these taxa. Sequencing reads from the metatranscriptomes were aligned to the genomes and plasmids of M. nitratireducenticrescens, and of Hyphomicrobium nitrativorans. Between 2.7% to 8.5% of reads aligned to these genomes from the five metatranscriptomes (Data S4). Reads that did not align with these genomes were de novo assembled, and the relative transcript levels of the resulting contigs (expressed as TPM-RSEM) was determined by aligning reads to them. Contigs were analyzed for the presence of ORF and processed into databases to determine their putative function and their most likely taxonomic affiliation (Data S4).

We identified nine-three taxa across the five biofilm samples, of which sixty-three were found in all samples. The proportion of transcripts affiliated to Methylophaga spp. (excluding nitratireducenticrescens) accounted for approximately 49% in the Original Biofilm RNA sample but decreased considerably (0.19 to 1.3%) in the RR biofilm RNA samples (Table 5). Seventeen taxa in low proportion in the Original Biofilm RNA sample showed substantial increases (one to four orders of magnitude) in their relative transcript levels in at least one of the RR biofilm RNA samples, while those affiliated to the genera Sedimenticola, Dechloromarinus, Cycloclaticus and Leptolyngbya showed decreases of one to three orders of magnitude in all RR biofilm RNA samples (Table 5). The relative levels of transcripts affiliated to the genera Bradymonas, Roseovarius and Cand. Promineofilum remained in the same proportion (within an order of magnitude) in all biofilm RNA samples (Table 5).

Table 5 Taxonomic affiliation of contigs with ORF and their proportions.

	OB	C1-1	C4-1	C6-1	C8-1	
Bacteroidota						
Bacteroidales						
Lentimicrobium	<0.1	0.61	1.3	4.8	1.1	
Marinilabiliales						
Geofilum	0	2.5	2.2	0.88	1.1	
Mariniphaga	<0.1	0.21	2.0	0.85	0.20	
Sunxiuqinia	<0.1	1.1	2.1	5.0	0.71	
Flavobacteriales						
Aequorivita	0.1	0.89	2.6	0.46	9.1	
Muricauda (Flagellimonas)	<0.1	34.1	4.0	9.5	14.1	
Pseudomonadota; Alphaproteobacteria						
Hyphomicrobiales						
Cohaesibacter	0.19	0.33	0.11	2.1	<0.1	
other Hyphomicrobium	0.71	0.36	1.7	1.3	7.5	
Stappia	<0.1	4.4	8.2	14.6	12.4	
Rhodobacterales						
Paracoccus	<0.1	1.7	5.0	3.9	2.8	
Roseovarius	0.44	0.42	0.48	1.2	0.39	
Rhodospirillales						
Oceanibaculum	1.4	7.0	8.0	8.7	23.4	
Pseudomonadota; Gammaproteobacteria						
Alteromonadales						
Idiomarina	0.11	11.7	3.6	2.2	1.5	
Chromatiales						
Sedimenticola	8.4	<0.1	<0.1	<0.1	<0.1	
incertae sedis						
Dechloromarinus	4.0	<0.1	<0.1	<0.1	<0.1	
Oceanospirillales						
Marinicella	0.10	1.1	9.3	0.23	0.60	
Pseudomonadales						
Marinobacter	0.39	12.1	25.0	14.3	3.8	
Thiotrichales						
other Methylophaga	48.7	0.19	0.33	1.3	0.38	
Cycloclasticus	2.7	<0.1	<0.1	<0.1	<0.1	
Deltaproteobacteria; Bradymonadales						
Bradymonas	3.5	4.0	2.9	0.82	1.7	
Spirochaetota; Spirochaetales						
Spirochaeta	<0.1	2.3	4.0	3.7	1.9	
Chloroflexota						
Cand. Promineofilum	3.1	0.26	0.89	2.2	0.87	
Cyanobacteriota; Leptolyngbyales						
Leptolyngbya	7.1	<0.1	<0.1	<0.1	0.22	
Mycoplasmatota						
Tenericutes	0	3.2	2.6	1.8	0.48	
Thermodesulfobacteriota; Desulfovibrionales						
Pseudodesulfovibrio	<0.1	0.95	0.47	2.9	0.36	
Others (67)	18.6	6.0	10.5	13.6	9.1	
Notes.

Assembled transcript reads (Contigs) containing ORF were grouped based on their affiliation. TPM-RSEM associated to these ORF were added and the proportion (%) derived.

Others: proportion of TPM-RSEM of the 67 other taxa.

Functional diversity

In addition to the microbial diversity provided by the metatranscriptome data, we can infer the potential metabolic pathways expressed in the microbial taxa active in the biofilm samples. Therefore, contigs with ORF that clustered according to their taxonomic affiliation provided an estimate of the metabolic activities occurring in the associated taxa in the biofilm. Among the ninety-three identified taxa, twenty-five taxa, in which their relative transcript levels of contigs were >2% TPM-RSEM in at least one RNA sample, were examined for putative metabolic pathways expressed in these taxa (Table 5; Data S6). Genes involved in denitrification (the four reductases) and in the carbon cycle (Fig. 6) were further examined as these pathways could explain the potential of these taxa to harvest energy and to assimilate the C1 carbon for their growth and maintenance. The bacterial pathways chosen were as follows.

Figure 6 Carbon assimilation and dissimilation pathways of methanol and formaldehyde.

Pathways were retrieved from KEGG and MetaCyc databases. Dashed arrows refer to pathways with multiple steps. C2, C3, C4, etc. refer to the number of carbons of the corresponding molecules. MeTHF: Methylene tetrahydrofolate. Ery4P: Erythrose-4P. Ru5P: Ribulose-5P. Ru1,5P: Ribulose-1,5P. Xu5P: Xylulose-5P. Ri5P: Ribose-5P. Sedoh7P: Sedoheptulose-7P. G6P: Glucose-6P. F6P: Fructose-6P. Pyr: Pyruvate. Glyc: Glycerate. Gluc6P: Gluconate-6P. OAA: Oxaloacetate. OHPyr: Hydroxypyruvate. Ac-CoA: Acetyl-CoA. shmt: Serine hydroxymethyltransferase. PEP: Phosphoenolpyruvate carboxylase. THMPT: Tetrahydromethanopterin. Glut: Glutathione. Mxa: Methanol dehydrogenase. XoxF: lanthanide [Ln 3+]-dependent methanol dehydrogenase. CODH: carbon monoxide dehydrogenase. Cyt: Cytochrome. PycAB: pyruvate carboxylase. CynT: carbonic anhydrase. Dissimilatory pathways are blue shaded.

1-Methanol oxidation to formaldehyde by the methanol dehydrogenase (Mdh), which provides gain of reductive equivalents by reducing cytochromes for the respiratory chain (Anthony, 1982).

2-Formaldehyde dissimilation pathways to formate by formaldehyde dehydrogenases (Fad; glutathione-dependent or independent) or by the tetrahydromethanopterin (THMPT) pathway, and formate to CO2 by formate dehydrogenases (Fdh). Gain of reductive equivalents (NADH) was obtained in these dissimilation pathways (Anthony, 1982; Vorholt, 2002).

3-Formaldehyde assimilation pathways through the ribulose monophosphate (RuMP). Associated with RuMP is the Entner–Doudoroff pathway, with carbon assimilation to pyruvate and glyceraldehyde3P, or carbon dissimilation by regenerating Ru5P with a lost of CO2 but gain of energy (NADPH) (Anthony, 1982; Vorholt, 2002).

4-Formaldehyde assimilation pathway through the methyl transfer (spontaneous) to tetrahydrofolate (THF) generating methylene-THF (meTHF), which is in turn incorporated in the biomass by the serine pathway or putatively by the reverse glycine cleavage system (RevGCS) (Bar-Even et al., 2013). Formate can be transformed to meTHF by the THF pathway (Anthony, 1982).

5-The Calvin cycle and the ethylmalonyl-CoA pathway (Alber, 2011) were also examined as associated with the pentose pathway or the Serine pathway. These pathways can provide additional carbon in the biomass through CO2 incorporation (Anthony, 1982).

Nineteen taxa contain transcripts associated with denitrification genes (Table 6), among which genes encoding all four reductases were found in the genera Hyphomicrobium, Marinobacter, Methylophaga, Sedimenticola and Stappia. Besides Methylophaga spp. and Hyphomicrobium spp., Paracoccus spp. was the only other taxon with transcripts associated with methanol dehydrogenase (Table 6). The formaldehyde dissimilation pathways (Fad, THMPT, Fdh; Fig. 6) were found in eighteen taxa, among which three have THMPT and Fdh, and eight have Fad and Fdh (none have the three; Table 6). Only Methylophaga spp. use the RuMP pathway (Fig. 6). The THF pathway and the serine pathway (Fig. 6) were found in 11 and 10 taxa, respectively (6 taxa contain both; Table 6), while GCS was found in all taxa. The Calvin cycle and ethylmalonyl-CoA pathway (Fig. 6) were found in six and four taxa, respectively (Table 6).

Table 6 Transcripts associated with genes involved in denitrification, and in formaldehyde assimilation and dissimilation pathways.

			Dissimilation	Assimilation	
Taxon	Denitrification	Mdh	Fad	THMPT	Fdh	THF	RuMP	Calvin	Ser	EM	ED	
Aequorivita *	nir, nor, nos	-	-	-	-	+	-	-	-	-	-	
Bradymonas *	nap, nir	-	+	-	+	-	-	-	+	-	-	
Cand. Promineofilum	nir, nos	-	-	-	+	-	-	-	-	-	-	
Cohaesibacter	nir, nor, nos	-	+	-	+	-	-	-	-	-	-	
Cycloclasticus *		-	+	-	+	-	-	-	+	-	-	
Dechloromarinus *	nar, nor, nos	-	+	-	+	-	-	+	+	-	-	
Geofilum	nor	-	-	-	-	+	-	-	-	-	-	
Hyphomicrobium	nap, nir, nor, nos	+	-	+	+	+	-	-	+	+	-	
Idiomarina *	nar, nir, nor	-	+	-	-	-	-	-	-	-	-	
Lentimicrobium		-	-	-	+	-	-	-	-	-	-	
Leptolyngbya *		-	+	-	-	-	-	+	-	-	-	
Marinicella	nir, nor	-	+	-	-	-	-	-	-	-	-	
Mariniphaga		-	-	-	-	-	-	-	-	-	-	
Marinobacter *	nar, nir, nor, nos	-	+	-	+	-	-	-	+	-	+	
Methylophaga	nar, nir, nor, nos	+	-	+	+	+	+	+	-	-	+	
Muricauda (Flagellimonas)	nor, nos	-	-	-	-	+	-	-	-	-	-	
Oceanibaculum	nar	-	+	-	+	+	-	-	+	-	-	
Paracoccus	nar, nir, nor, nos	+	+	-	+	+	-	+	+	+	+	
Pseudodesulfovibrio *	nar	-	-	-	+	-	-	+	-	-	-	
Roseovarius *	nir, nor, nos	-	+	-	+	+	-	-	+	+	-	
Sedimenticola *	nar, nir, nor, nos	-	-	+	+	+	-	+	+	-	-	
Spirochaeta *		-	-	-	-	-	-	-	-	-	-	
Stappia *	nap, nir, nor, nos	-	+	+	+	-	-	+	+	+	
Sunxiuqinia	nir, nor	-	-	-	-	+	-	-	-	-	-	
Tenericutes		-	-	-	-	-	-	-	-	-	-	
Notes.

* Pathways were identified by the KEGG mapper reconstruct analysis with data from the metatranscriptomes, and by the KEGG pathways analysis with a strain representative of the taxon when available.

+: presence; -: absence.

See Fig. 6 for pathway description.

Mdh, Methanol dehydrogenase (mxa mainly); Fad, Formaldehyde to formate by the formaldehyde dehydrogenase (either glutathione-dependent or -independent); THMPT, Process of formaldehyde to formate through the Tetrahydromethanopterin (THMPT) pathway; Fdh, Formate dehydrogenase; THF, Process of formate to Methylene tetrahydrofolate (meTHF) through the THF pathway; RuMP, Formaldehyde assimilation by the Ribulose-5P pathway, Calvin, Calvin cycle here is defined as taxon containing transcripts encoding the phosphoribulokinase (Prk) and the ribulose-1,5-bisphosphate carboxylase/oxygenase (rubisco) enzymes; Ser, Formaldehyde assimilation by the serine pathway; EM, Ethylmalonyl-CoA pathway; All taxa have complete pathways of glycine cleavage system (GCS), tricarboxylic acid cycle (TCA) and glycolysis; ED, Entner–Doudoroff pathway.

Discussion

C/N and the denitrification performance

Our results showed that different C/N did not have a significant impact on the reactor denitrification performance (NOx/NO3− reduction rates, NO2− and N2O dynamics), but did have an impact on the methanol consumption rates and the CO2 production rates. This impact may be linked to the initial concentrations of methanol (i-Methanol) applied to the reactor, as these concentrations also correlated with the methanol consumption rates and the CO2 production rates. Interestingly, the initial concentrations of NO3− (i-NO3−) did positively correlate with the denitrification performance, but such correlation was not observed with the i-Methanol. Our results suggest that opposite effects occur between the i-NO3− and the i-Methanol applied to the reactor, resulting in a null effect by C/N on the denitrification performance, but not on the carbon metabolism.

In methanol-fed denitrification processes, methylotrophic bacteria use methanol as source of carbon and energy, where it is first converted to formaldehyde by the methanol dehydrogenase with transfer of electrons to cytochromes and then to the respiratory chain. Formaldehyde can either be assimilated into the biomass, or dissimilated in formate and CO2 by enzymatic reactions generating NADH (Anthony, 1982). Under denitrifying conditions (anoxic), NO3− and N oxides replace O2 as terminal electron acceptors for respiration. NO3− acts as an electron acceptor with the NO3− reductase system (Nar system) contributing to the proton motive force, while NO2−, NO and N2O accept electrons from electron pools generated from the respiratory chain (Simon, 2011). Our results suggest that the level of NO3− reached a saturation point where the denitrification performance could not increase. This was not the case with methanol, where it can contribute to both energy and cell growth and maintenance. Therefore, as the methanol concentration increased, higher methanol consumption (input) and CO2 production (output) occurred without the need for greater denitrifying activities.

The reactor was able to reduce most, if not all, all i-NO3− applied to the reactor after four operating days, although the higher concentrations (32.1 and 42.8 mM) generated higher levels of residual NO2−, which would be eventually reduced over time. The reactor did not generate ammonium, and negligible amount of N2O was found at the end of the reaction phase. However, high proportions of residual methanol (47 to 91%) were found at the end of the reaction phase, which is not a desired outcome for a denitrification system. The release of methanol in the effluent of a water treatment can be harmful to the ecosystem. For example, Pungrasmi, Playchoom & Powtongsook (2013) showed that applying a C/N of 4 with methanol in their denitrification system resulted in the best denitrification performance. However, due to the concentration of residual methanol exceeded toxic level for fish development, a C/N of 3.3 had to be applied in their system when treating the effluent from a recirculation (tilapia) aquaculture system (RAS).

We acknowledge that analysing a single reactor limited the significance of our findings. Running multiple reactors in parallel is challenging to maintain synchronization of conditions and can be a burden on human resources, as the reactors operate continuously and require daily attention. The initial strategy was to operate all eight conditions three times (three periods) with the same biofilm, but due to the COVID crisis, only two could be performed. We noticed however that, as the biofilm ages, the microbial community evolved from the early stages, and the biofilm thickens with active and dead zones, increasing the variability of the operating conditions. Ideally, each condition should have been conducted with fresh acclimated biofilm in triplicate (i.e., eight reactors with fresh acclimated biofilm, in triplicate).

Evolution of the bacterial community

Both approaches (16S amplicon sequencing and metatranscriptomes) to monitor the evolution of the bacterial diversity profiles in the biofilm during the operating conditions showed that the bacterial community established in the reactor diverged substantially from the Original Biofilm collected in the Biodome denitrification system in the proportion of the common taxa. These changes can be explained by the operating conditions: laboratory-scale sequential batch-mode recirculating reactor versus full-scale continuous operating mode in the Biodome denitrification system. Although both processes used the same medium (IO), the input water in the Biodome system contained additional matter from animal and plant waste, or non-ingested food (Parent, Labbé & Villemur, 2002), and additional microorganisms from the aquatic ecosystem (e.g., protozoa) (Laurin et al., 2008) that might have influenced its microbial community.

The results obtained with 16S amplicon sequencing revealed that a high proportion of Methylophaga spp. established in the recirculating reactor during the acclimation phase (D0), and that subsequently the biofilm populations increased in diversity at the expense of Methylophaga spp. (although still constituting a high proportion of the bacterial community), in favor of heterotrophic bacteria such as the genera Stappia, Marinobacter, Marinicella, Oceanibaculum and Paracoccus. Methylophaga spp., Hyphomicrobium spp., Paracoccus spp., Marinobacter spp. and Marinicella spp. have been found in other marine denitrification systems (Güven, 2009; Lu, Chandran & Stensel, 2014; Furukawa et al., 2016; Wang et al., 2023). Some members of these taxa have denitrification capacity such as Stappia spp., Paracoccus spp. and Marinobacter spp. Contrary to what was stated by Duc et al. (2018) and echoed by other reports about Shoji et al. (2014), the sulfur oxidizing bacteria Marinicella were not shown to carry NO3− reduction to N2 in this report. Results from metatranscriptomes, however, revealed that nirK and nor genes were found to be affiliated to Marinicella spp. None of the Marinicella sp. sequenced genomes in GenBank contain the full set of denitrification reductases. Stappia spp. are marine chemoorganotrophic bacteria that can oxidize CO, and some species contain a gene for the large subunit of ribulose-1,5-bisphosphate carboxylase/oxygenase (RuBisCO) (cbbL) that may suggest the ability to couple CO utilization to CO2 fixation (Weber & King, 2007).

Potential metabolic pathways enabling the transformation of formaldehyde for carbon assimilation or dissimilation (Anthony, 1982; Vorholt, 2002; Yurimoto, Kato & Sakai, 2005) could explain the persistence and growth of heterotrophs in the reactor, as these pathways were found in the observed taxa. For instances, the oxidation of formaldehyde and formate generates NADH by their respective dehydrogenases. Formaldehyde is produced by the action of the methanol dehydrogenase, carried by Methylophaga spp., Paracoccus spp. and Hyphomicrobium spp., at the periplasm (Anthony, 2004). Due to high proportion of Methylophaga spp. and high level of methanol, it is possible that excess of formaldehyde diffused into the biofilm and was available to the other organisms because of their promiscuity (Chistoserdova & Kalyuzhnaya, 2018). The emergence of these heterotrophic bacteria could also be the result of subsequent colonization of the biofilm after the early colonization of the supports by the methylotrophs such as Methylophaga spp., providing complex carbon to the heterotrophs such as carbohydrates in the extracellular polymeric substances of the biofilm matrix or organic waste from dead cells as the biofilm aged.

Relative expression profile of strain GP59

M. nitratireducenticrescens strain GP59 has adapted to the recirculating reactor resulting in colonization of supports in high proportion. Following the early stage of colonization, heterotrophs increased in proportion during the operating conditions. Regarding the relative transcript levels of strain GP59, these were influenced by the physiology state (biofilm vs planktonic), the culture media, and the operating conditions (oxic vs anoxic).

High proportions (22.5 to 68.9%) of 16S rRNA gene sequences affiliated with Methylophaga spp. were found in all RR biofilm samples, six to 18 times higher than in the Original Biofilm (3.7%). However, despite this low level in the Original Biofilm, transcript reads associated with strain GP59 and other Methylophaga species in the Original Biofilm together accounted for approximately 50% of the overall relative transcript profile of the Original Biofilm, suggesting that Methylophaga spp. were more active in the Original Biofilm than the other taxa. For example, transcript reads associated with Hyphomicrobium spp. in the Original Biofilm RNA sample represented <1% relative transcript levels, whereas it represented >40% in 16S rRNA gene sequences in the DNA sample. Although relative transcript levels of strain GP59 had 4-fold increases on average in the RR biofilm RNA samples, these levels associated with the other Methylophaga species were almost negligible, suggesting that strain GP59 has better adapted to the anoxic recirculating conditions than the other Methylophaga species. The denitrification system at the Montreal Biodome, although containing a deoxygenation reactor, was not entirely airtight and might have sufficient O2 (0.3–0.8 mg O2/L; Labbé et al., 2003) for the development of obligate aerobic Methylophaga sp. such as M. frappieri strain JAM7 that we isolated from this system (Auclair et al., 2010).

The relative gene expression profiles of strain GP59 in the RR biofilm were clearly different from those observed in (i) the Original Biofilm, (ii) the ASW-medium, batch-static biofilm cultures, and (iii) the planktonic cultures with the specific Methylophaga medium (Methylophaga-1403). These results strongly suggest that the medium, the physiological conditions (planktonic, biofilm) and the operating conditions (anoxic, oxic) have a significant impact on the overall gene expression profiles in strain GP59 (Di Capua et al., 2022).

The relative expression profiles of genes involved in N-cycle in strain GP59 had a similar pattern in the four RR biofilm samples and in the PlkAN cultures, suggesting that the operating conditions (planktonic cultures versus recirculating biofilm reactor) did not modify the expression profiles of strain GP59 regarding the N-cycle despite changes in C/N in the recirculating reactor. The relative expression patterns of the gene clusters of the nar1 and nor1 systems and genes involved in NO response appear to have been more engaged under these conditions, unlike the nar2 and nor2 systems, which were more involved under the oxic conditions or in the Original Biofilm. We demonstrated in a previous study (Lestin & Villemur, 2024) that nar1 (Nar: NO3− reductase) had higher expression levels in strain GP59 planktonic cultures than those of nar2, suggesting higher denitrifying activities in the RR biofilm and the PlkAN cultures. These higher activities would generate higher levels of NO, explaining the higher relative transcript levels of genes involved in NO response.

All four xoxF genes had higher relative transcript levels in the four RR biofilm samples. The XoxF-type lanthanide (Ln3+)-dependent methanol dehydrogenases are widespread among bacteria. Ln3+ is an important co-factor that regulates the expression of the xoxF and the mxa gene cluster (encoding Ca2+-dependent methanol dehydrogenase) (Chistoserdova & Kalyuzhnaya, 2018). We do not know whether Ln3+ was included in the IO commercial formulation, or whether it exists naturally as trace element. However, the fact that the xoxF genes had higher relative transcript levels in the RR biofilm samples in contrast to those of mxa which were higher in the oxic planktonic cultures, suggests important roles of XoxF in the carbon dynamics in biofilm under anoxic conditions as alternative methanol dehydrogenases (Chistoserdova & Kalyuzhnaya, 2018). Mustakhimov et al. (2013) proposed a connection between XoxF in providing electrons from methanol oxidation and the denitrification pathway in the methylotrophic Methylotenera mobilis JLW8.

Along with the xoxF expression patterns, similar results were observed with 10 of the 11 riboswitches with higher relative transcript levels in the RR biofilm samples. Examination of the genes flanking the eleven riboswitches showed diverse functions. Four of them (Ribo4, 5, 6, 7; c-di-GMP type) are located in close proximity (18 kb apart), and each flanked a gene encoding a protein with PEP-CTERM domain, which appears to be involved in exopolysaccharide-associated protein sorting systems (Haft et al., 2006; Villemur et al., 2019). Furthermore, genes surrounding these coding sequences encode proteins involved in exopolysaccharide biosynthesis and export. In Gram-negative bacteria, many synthase-dependent exopolysaccharide secretion systems are post-translationally regulated by an inner-membrane c-di-GMP receptor (Whitney & Howell, 2013). Interestingly, Ribo5 had higher relative transcript levels in planktonic anoxic cultures (PlkAN), while the three others had higher levels in the RR biofilm samples. These results suggest that specific controls of these pathways affecting the bacterial surface are important for the fate of the bacterial physiology (biofilm vs planktonic). It is known that c-di-GMP is a key factor in bacteria to regulate several genes involved in the planktonic-biofilm transition. One of the mechanisms of biofilm development is the surface attachment, which requires specific adherent organelles such flagella, fimbriae or pili (Martínez & Vadyvaloo, 2014).

Conclusions

C/N did not impact the denitrifying activities of the recirculation reactor but did impact the carbon dynamics, as higher C/N correlated with higher methanol consumption and CO2 production rates. The bacterial diversity profiles in the RR biofilm samples were different from the one of the Original Biofilm. From Day 0, as the RR biofilm aged, an increase in the bacterial diversity was observed, particularly in heterotrophic bacteria. The functional diversity profiles suggest that heterotrophs may use formaldehyde, which could have been released by methylotrophs in the biofilm, as energy and carbon source. The relative gene expression profiles of M. nitratireducenticrescens strain GP59 in the RR biofilm samples were distinct from those derived from the strain GP59 planktonic pure cultures and those from the Original Biofilm, suggesting that the operating conditions and the culture media are key factors influencing these expression profiles.

Our results suggest that large-scale denitrification systems would benefit from long-term operation, as the aging microbial community evolves into more diverse populations capable of withstanding a wide range of C/N variations. This type of microbial community can also adapt more readily to changes in the carbon source (C-1 carbon to complex carbons or mixture of both). For instances, in aquaculture, fish feed and excreta can add various carbon sources and generate higher levels of NO3− in water, which can disturb the C/N parameters in the denitrification system. Therefore, adapting the microbial community to its new environment will be feasible and should lead to the establishment of a stable and efficient denitrification system. We believe that knowing the evolution of the biofilm microbial community under the operating conditions is an important outcome of our research. It will certainly help to understand the microbiology of these bioprocesses and to understand what is behind all these technical configurations.

Supplemental Information

Supplemental Information 1 Biofilm on the reactor supports

Supplemental Information 2 Physico-Chemical measurements

This file describes the analysis of the raw data of the measured physical-chemistry parameters that include (1) the nitrate and NOx reduction rates, (2) the nitrite and N2O dynamics, (3) the methanol consumption rates, and (4) the CO2 production rates for the eight conditions and the two periods. Also provided, the global results and the Pearson correlation analysis.

Supplemental Information 3 Accession numbers

Supplemental Information 4 16S diversity profiles

Supplemental Information 5 Metatranscriptomes

Supplemental Information 6 GP59-Multivariate analysis

Supplemental Information 7 Functional taxa

Supplemental Information 8 Riboswitch coordinates and relative transcript levels

We thanks Professeur Philippe Constant of the Institut national de la recherche scientifique for performing PERMANOVA.

Additional Information and Declarations

Competing Interests

Author Contributions

Data Availability

The authors declare there are no competing interests.

Livie Lestin conceived and designed the experiments, performed the experiments, analyzed the data, prepared figures and/or tables, authored or reviewed drafts of the article, and approved the final draft.

Richard Villemur conceived and designed the experiments, analyzed the data, prepared figures and/or tables, authored or reviewed drafts of the article, and approved the final draft.

The following information was supplied regarding data availability:

Raw sequencing data are available in the Sequence Read Archive (SRA): PRJNA1194547, SAMN45188687 to SAMN45188690, PRJNA744510, SAMN20104466, PRJNA525230, SAMN11029462, SAMN11029465, SAMN11029466, PRJNA1072961, SAMN39755719 to SAMN39755725, PRJNA525230, SAMN11043171 to SAMN11043173, PRJNA1194547, SAMN45195935 to SAMN45195938, PRJNA524642, SAMN11029470.

Assembled contigs from metatranscriptomes are available at the Joint Genome Institute IMG/MER: Ga0510006 to Ga0510010.

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
