# Peer review of "Modulation of carbon-to-nitrogen ratio shapes the microbial ecology in a methanol-fed recirculating marine denitrifying reactor"

_PeerJ, doi:10.7717/peerj.20129_

## Round 0.1 · original submission · Major Revisions

Please address all the reviewers' concerns.

Reviewer 1 ·

Basic reporting

Title & Abstract
Title: The Title is too long, descriptive, and does not summarize the prime findings of the study. Consider revising the Title as follows: “C/N ratio modulation shapes the microbial ecology in a recirculating marine denitrifying reactor.”
Abstract: The relevance and significance of the study are not clarified in the Abstract. Also, the prime objectives of the study are not defined. Hence, the Abstract can be revised according to the following suggestions to highlight the prime findings of the study.
• Lines 21–26: The authors state that “Carbon-to-nitrogen ratio (C/N) is an essential parameter known to influence….”; however, it is the associated issues with this parameter need to be addressed. Also, the need for this study should be justified to contribute to the field rather than only adding to the existing knowledge.
• Line 29: The relevance for eight different C/N ratios and 31 days of operation needs to be specified. Please provide a relevance statement.
• Please specify the ratios of C/N used in the study.
• Lines 32–35: “Metatranscriptomes representative conditions were performed to…” – Please specify the representative conditions. Also, the different stages of the biofilm at which the analysis was performed should be specified.
• Lines 36–44: The results are simple statements that are not supported by any statistical analysis.
• Line 33: “Changes in C/N did not impact the denitrifying activities of the recirculating reactor but
the carbon dynamics.” – Please provide supporting data for the statement. How did the authors determine that the C/N ratio had an impact on the carbon dynamics?
• Lines 38–40: “The bacterial community in the reactor increased in diversity as the biofilm aged….”
It is unclear whether the changes in C/N ratio influenced the bacterial community. Please explain the role of changes in C/N ratio in bacterial ecology.
• Line 40: Was similar functional diversity observed in the biofilms formed for all the C/N ratios tested?

Introduction
The Introduction is descriptive. In the current format, the need for the study is not described precisely. Also, the background information does not correlate with the Methods used to fulfill the stated objective.
Thus, the section needs to be revised to enhance the significance of the study based on the following suggestions:
• Elaborate the issues associated with the denitrification of wastewater.
• Explain why the process needs to be optimized depending on the source of the pollutant.
• Specify why marine denitrification needs to be addressed.
• Describe the role of microbial composition in the denitrification process.
• Discuss the microbial denitrification and ecological implications.
• Please discuss how the changes in C/N ratio could affect the microbial ecology and how these influence the efficacy of the denitrification.
• Justify why methylotrophic and marine conditions need to be studied for denitrification.
• How does 16S RNA sequencing help to assess the microbial diversity and provide knowledge on the denitrification process?
• Justify the usage of metatranscriptomics in understanding the functional diversity as influenced by various environmental conditions.
• Define the objectives.
• Line 131: “….M. nitratireducenticrescens strain JAM1 and H. nitrativorans strain NL23…” Please clarify the relevance of the usage of the two strains.
Consider including the following references to support the claims of the missing information as per the above suggestions:
1. Choi Y, Ryu J, Lee SR. Influence of carbon type and carbon to nitrogen ratio on the biochemical methane potential, pH, and ammonia nitrogen in anaerobic digestion. J Anim Sci Technol. 2020 Jan;62(1):74-83. doi: 10.5187/jast.2020.62.1.74. Epub 2020 Jan 31. PMID: 32082601; PMCID: PMC7008128.
2. Feng Y, Wang L, Yin Z, Cui Z, Qu K, Wang D, Wang Z, Zhu S, Cui H. Comparative investigation on heterotrophic denitrification driven by different biodegradable polymers for nitrate removal in mariculture wastewater: Organic carbon release, denitrification performance, and microbial community. Front Microbiol. 2023 Feb 20;14:1141362. doi: 10.3389/fmicb.2023.1141362. PMID: 36891393; PMCID: PMC9986267.

Figures & Tables
The Figures and Tables are adequate.
The Figures are readable and supported by an adequate amount of data. However, please specify the statistical tools employed along with the level of significance in the legends of the Figure and footnotes of the Tables.

Experimental design

Material and Methods
The Methods section is inadequate. Hence, additional information is requested for data reproducibility based on the following suggestions:
• Lines 159-160: Did the authors perform any viability assays to estimate the total microbial load in the biofilm before and after the transport? If yes, specify and describe the assay; if not, justify the reasons.
• Lines 162-163: The inoculum was adapted to 4 oC; please explain the relevance.
• Line 192: “the running operation was abruptly interrupted…” – Specify whether the interruption affected microbial viability and influenced microbial community shifts.
• Lines 186–215: Specify the sampling frequency and whether the samples were immediately processed for DNA or RNA isolation.
• Specify the quality filtering thresholds for dada2 pipeline.
• Lines 241-–74: Specify the number of biological and technical replicates.
• Expand the information on the bioinformatics workflow, including rRNA depletion, gene annotation, normalization, etc.
• Specify the taxonomic assignment methods employed and describe the criteria for taxon-function grouping.
• Include a paragraph on statistical analysis of the data.

Validity of the findings

Results
The Results section is descriptive, and the findings are correlated with the data. However, it can be modified further based on the following suggestions to enhance the significance of the study.
• Line 357: “It revealed that changes in C/N did not correlate with the denitrification dynamics..” – Typically, the changes in the C/N ratio alter the electron flow and denitrification rates. Please discuss this observation.
• Provide the time-course profiles for NO₃⁻, NO₂⁻, and N₂O.
• Correlate the riboswitch expression data with the adjacent gene transcription patterns, which will provide insight into the changes in metabolic profile.
• Provide statistical significance values for all comparative analysis data.

Discussion
The Discussion is provided in several sub-sections; however, for better understanding, it is recommended to present an unstructured Discussion for a single continuous story and incorporate the missing information based on the following suggestions.
• Reiterate the prime findings of the study in the first paragraph.
• Line 192: The authors mention that the running operation was abruptly interrupted. Please discuss whether the interruption affects the microbial viability and if it has any effect on the microbial community shifts.
• Discuss the limitations of 16S approach and the primer bias while describing the microbial communities.
• Discuss the expression levels of specific genes involved in denitrification with reactors’ denitrification performance, N₂O production, or methanol consumption.
• Line 357: The authors made an observation indicating that the changes in C/N did not correlate with the denitrification dynamics. Typically, the changes in the C/N ratio alter the electron flow and denitrification rates. Although a short explanation is provided in lines 560–564, further elaboration with specific citations is essential to establish the correlation with the literature.
• Please discuss the limitations of the study.

Conclusion
The Conclusion is not supported by adequate data. The authors have found a correlation between C/N ratio, denitrification rate, and microbial diversity. However, no data show that the shift in microbial community heterogeneity plays an explicit role in the efficacy of the denitrification events.
Also, due to the lack of biological replicates, it is difficult to understand that a specific C/N ratio is associated with a particular microbial community structure. Therefore, sentences in lines 701–704 can be removed for specificity rather than speculation.

Reviewer 2 ·

Basic reporting

no comment

Experimental design

no comment

Validity of the findings

no comment

Additional comments

The manuscript explores the effects of varying C/N ratios in a methanol-fed denitrification reactor under marine recirculating conditions. The study also provide insights into microbial community shifts and gene expression patterns using metatranscriptomic analyses. While the topic is novel and promising, the execution and presentation of the work lack rigor, and the overall reproducibility and clarity of the experimental design are insufficient. Key findings related to C/N ratio effects are not adequately substantiated. Below are specific comments:

1. Lines 107–110: Please update the literature examples using more recent sources. It would also be helpful to specify whether methanol was used as the carbon source in each study, to provide stronger context.
2. Regarding Phase II: Please clarify whether the same original biofilm was reused from Phase I and whether it underwent the same inoculation and adaptation protocol. Otherwise, it may not be appropriate to state that each step was repeated.
3. Since nitrate concentrations were adjusted using NaNO₃, the resulting variation in Na⁺ concentrations could affect microbial activity. Please comment on whether this was considered and whether it had any noticeable impact.
4. Lines 175–176: The authors mention N₂ sparging to maintain anoxic conditions. Please clarify whether this N₂ flow was considered in the gas production measurements.
5. Line 180: Please define what is meant by "denitrifying activity." Does this refer to nitrate removal, nitrogen gas production, or another specific metric?
6. It appears that each C/N condition was only tested once. If this is the case, please explain the reasoning for not performing replicates or repeated batches, as this raises concerns about microbial community stabilization and data reproducibility.
7. Line 321: Please confirm whether using italics for "Original Biofilm" is appropriate. It is not a scientific name and may not require italicization.
8. Line 323: What was the rationale for selecting inoculum derived from the C/N 3 condition? Additional justification would strengthen the methodological transparency.
9. Line 349: Please correct unit formatting: "10h to 20h" should be revised to "10 h to 20 h."
10. Section 3.3: The choice to analyze only C5-1, C5-2, and C6-2 samples for community composition lacks justification. Please explain how these samples were selected and why they represent the broader C/N conditions.
11. Line 464: This should read "0% NaCl." Please ensure consistent formatting throughout the manuscript.
12. Please present the gene expression data in a quantitative. If the data are predicted rather than directly measured, consider whether the term "expression" may be an overinterpretation and revise accordingly.
13. Section 3.4.2: Please explain the criteria for selecting only C1-1, C4-1, C6-1, and C8-1 samples for transcriptional diversity analysis.
14. Section 3.4.2: Adding a short summary of the main conclusions would help reinforce the value of this section.
15. Section 3.4.3: Similarly, include a concise summary of key findings at the end of the section.
16. Section 4.1: The discussion in this section is vague. While it acknowledges the importance of C/N and operational parameters such as HRT, the take-home message is unclear. Please clarify what the authors ultimately conclude here.
17. Since the results show community shifts when the original biofilm was applied to the reactor, please explain whether using an inoculum already similar to the reactor’s established community would be more beneficial for system performance.
18. The manuscript suggests microbial diversity is important, but does not provide evidence to support this. Please clarify whether and why greater diversity is expected to improve reactor performance.
19. The conclusion section reads more like a set of suggestions. The core conclusions related to the effects of C/N ratio should be presented more prominently.
20. Table 1: Please add parentheses (e.g., "(h)") to the column headings under P1 and P2. If the values in parentheses represent error margins, this should be specified in the caption.

---

## Round 0.2 · Minor Revisions

Please address the remaining comments.

Reviewer 2 ·

Basic reporting

no comment

Experimental design

no comment

Validity of the findings

no comment

Additional comments

no comment

·

Basic reporting

The report text is written in a completely understandable way. The description of the problem is very clear ans redacted in great detail.
The information obtained and explained in the conclusions, is of gear value for the health of commercial and educational (aquariums) aquatic systems; even home aquariums could have benefits from this research.

Experimental design

The main objective is to point out how the C/N radio affects the denitrification function in a closed system by the bacterial community, and the role of M. nitratireducenticrescens strain GP59 as the trigger for the correct establishment and function of such bacterial community. The experiments designed here, respond to that points.

Validity of the findings

As mentioned, the importance of nitrogen removal from the aquatic systems is fundamental for the health of other macro species, animal species, specifically. The findings here, are helping to understand the function of the bacterial community in charge of that task. It is relevant for large and small scale aquatic installations.

Additional comments

Just some corrections:
79. are composed mostly of methylotrophic bacteria (used C1 carbon as carbon source), which are
different from acetate-fed processes,
87. a higher proportion of the carbon source is used as electron donors for the denitrification (Ni et al., 2017).
125. It would be of important educational value to explain a bit further, about the origin of the denitrification biofilm, that one that comes from the Biodome. Is it lab made?? Can it be altered with new bacterial genera??
361. Why the biofilm was not stored at -80°C???
364. For descriptive objectives, use the same units.
421. although the M. nitratireducenticrescens band is more intense (higher proportion) in the 2004 sample

---

## Round 0.3 · accepted · Accept

Thanks for addressing all comments!

·

Basic reporting

Small redaction errors were corrected

Experimental design

No further requirements were asked for the experimental designs

Validity of the findings

The objectives of the work are covered

Additional comments

I have read and confirm the correction text with the author's responses. I found the in agreement with the questions made. My recommendation is to publish this work.